

**Daily gridded datasets of snow depth and snow water equivalent for the Iberian Peninsula from 1980 to 2014**

Esteban Alonso-González[1], J. Ignacio López-Moreno[1], Simon Gascoin[2], Matilde García-Valdecasas Ojeda[3], Alba Sanmiguel-Vallelado[1], Francisco Navarro-Serrano[1], Jesús Revuelto[4], Antonio Ceballos[5], María Jesús Esteban-Parra[3], Richard Essery[6].

[1] Instituto Pirenaico de Ecología, Consejo Superior de Investigaciones Científicas (IPE-CSIC), Zaragoza, Spain
[2] Centre d'Etudes Spatiales de la Biosphère (CESBIO), UPS/CNRS/IRD/CNES, Toulouse, France
[3] Departamento de Física Aplicada, Facultad de Ciencias, Universidad de Granada, Granada, Spain
[4] Météo-France - CNRS, CNRM (UMR3589), Centre d'Etudes de la Neige, Grenoble, France
[5] Dept. Geografía, Universidad de Salamanca, Salamanca, Spain.
[6] School of GeoSciences, University of Edinburgh, Edinburgh, UK

*Correspondence to*: Esteban Alonso-González (e.alonso@ipe.csic.com)

**Abstract**

We present snow observations and a validated daily gridded snowpack dataset that was simulated from downscaled reanalysis of data for the Iberian Peninsula. The Iberian Peninsula has long-lasting seasonal snowpacks in its different mountain ranges, and winter snowfalls occur in most of its area. However, there are only limited direct observations of snow depth (SD) and snow water equivalent (SWE), making it difficult to analyze snow dynamics and the spatiotemporal patterns of snowfall. We used meteorological data from downscaled reanalyses as input of a physically based snow energy balance model to simulate SWE and SD over the Iberian Peninsula from 1980 to 2014. More specifically, the ERA-Interim reanalysis was downscaled to 10 ×10 km resolution using the Weather Research and Forecasting (WRF) model. The WRF outputs were used directly, or as input to other submodels, to obtain data needed to drive the Factorial Snow Model (FSM). We used lapse-rate coefficients and hygrobarometric adjustments to simulate snow series at 100 m elevations bands for each 10 × 10 km grid cell in the Iberian Peninsula. The snow series were validated using data from MODIS satellite sensor and ground observations. The overall simulated snow series accurately reproduced the interannual variability of snowpack and the spatial variability of snow accumulation and melting, even in very complex topographic terrains. Thus, the presented dataset may be useful for many applications, including land management, hydrometeorological studies, phenology of flora and fauna, winter tourism and risk management. The data presented here are available for free download from Zenodo (DOI: 10.5281/zenodo.854618).This paper fully describes the work flow,



data validation, uncertainty assessment and possible applications and limitations of the
database.

## 1. Introduction

Seasonal snowpack exerts an important control on the hydrology and
economy,of many mountainous and cold regions worldwide (Barnett et al., 2005). Snow
variability also affects different ecological processes, such as species composition,
distribution, and phenology (Keller et al., 2000; Wipf et al., 2009). For example,
snowpack on Mediterranean mountains is a crucial source of water during the dry
season (Fayad et al., 2017; García-Ruiz et al., 2011; Viviroli et al., 2007). Long-term
data are required to analyze the spatiotemporal dynamics of snowpack, to assess the
importance of snow as resource, and understand the effect of climatic fluctuations.
However, there are only limited *in situ* observations of snowpack for most mountain
regions (Raleigh et al., 2016). Nowadays remote sensing techniques could only reliably
provide information about snow cover  based on observations in the visible spectrum
(Dietz et al., 2012). Current space-borne sensors do not provide accurate data on snow
water equivalent (SWE) and/or snow depth (SD) in mountainous regions (Dozier et al.,
2016). Microwave imaging has a coarse resolution (grid cell size: ~25 km), so does not
characterize snowpack variability in the Mediterranean mountains, which have a high
spatial heterogeneity not captured with this resolution. There are also spatial and
temporal limitations when attempting to estimate snowpack using close range remote
sensing techniques such as light detection and ranging (LIDAR) (Revuelto et al., 2016).

There are limited *in-situ* snow observations and meteorological data at high
elevations in the Iberian Peninsula. Although the number of monitored sites has
increased in recent years, there are no long-term series and there is insufficient
characterization of snowpack dynamics at a regional scale. However, snowpack in the
Iberian Peninsula is an important hydrological and also economical resource. An area of
19456.4 km$^2$ in the Iberian Peninsula lies above 1500 m.a.s.l., mostly in the five large
mountain ranges (Pyrenees, Cantabrian Mountains, Central System, Iberian Range and
Sierra Nevada). At this elevation, snowpack occurs for at least four months of the year
(López-Moreno et al., 2011) making it a critical resource for water management in the
largest hydrological basins (Morán-Tejeda et al., 2014). Snowpack influences the
interannual variability of water resources (López-Moreno and García-Ruiz, 2004) and



the timing of the winter low flows and spring peak flows (Sanmiguel-Vallelado et al., 2017). Moreover, winter tourism (mainly skiing) has an increasing importance to the
economy of mountain valleys during recent decades, and the large interannual fluctuations of snowpack in the different mountain regions of the Iberian Peninsula affect the economic viability of tourism (Gilaberte-Búrdalo et al., 2014, 2017).

     The importance of snow to the environment and economy of the Iberian Peninsula, and the lack of data on snowpack in this region, motivated us to use
meteorological outputs from downscaled reanalysis data to simulate snowpack at different elevations in the Iberian Peninsula. Atmospheric reanalyses, based on data assimilation and modeling (Saha et al., 2010), can provide important information about the temporal evolution of the atmosphere. Meteorological variables obtained from reanalysis data can be used as inputs for models of snow mass and energy balance
which can be applied to describe the behavior of the snowpack over large areas (Brun et al., 2013; Krogh et al., 2015; Wegmann et al., 2017). However, the coarse resolution (cell size: ~10s of km) implies these simulations may have insufficient spatial resolution for characterizing the topographical complexity of mountain areas (Mass et al., 2002). To overcome this limitation, Regional Climate Models (RCMs) are often used to obtain
better representations of surface climatology, because they downscale physically reanalysis products (García-Valdecasas Ojeda et al. 2017; Kryza et al. 2017; Warrach-Sagi et al. 2013). Previous studies have used RCMs to study SD and SWE dynamics at finer resolutions (grid cell size: 5 to 11 km) when they are driven with reanalyses, and the resolution increases further (grid cell size: 1 km) when using forecasted data
(Bellaire et al., 2011; van Pelt et al., 2016; Quéno et al., 2016; Wu et al., 2016).

     van Pelt et al. (2016) used the High Resolution Limited Area Model (HIRLAM) in Svalbard (Norway), with forcing by ERA-40 and ERA-Interim reanalysis, and then used the meteorological simulation as driving data for SnowModel (Liston et al., 2006a). Their results support the usefulness of the methodology extracting snowpack
trends from these data. Wu et al. (2016) used a similar procedure to describe the behavior of snowpack over the Altay Mountains in China. They coupled outputs from the Weather Research and Forecasting (WRF) model (Skamarock et al., 2008) driven by NCEP/NCAR reanalysis with a temperature index model (based on remote sensing), and their results had low error values. To increase the spatial resolution of the WRF
outputs, they used the MICROMET model (Liston et al., 2006b), a submodel of SnowModel in which WRF outputs are interpolated to a new grid, and then corrected



physically according to topography. Wrzesien et al., 2017 tested the capability of WRF to estimate SWE over complex terrain concluding that WRF simulations can be used over areas with few observational data.

We used a different approach, in an effort to make our database more computationally practicable and to avoid the uncertainties of the statistical interpolations of climatological variables over complex areas. More specifically, we projected WRF outputs to different elevation bands to generate simulations for multiple elevations. This procedure allows to study the elevation-dependent characteristics of the

snowpack over different mountain ranges, preserving the WRF output resolution.

     Our procedure uses the physically based Factorial Snow Model (FSM) (Essery, 2015), which is fed by ERA-Interim reanalysis (Berrisford et al., 2011), and downscaled by the WRF model. The final products of our analysis are simulated daily time series of SD and SWE at different elevations from 1980 to 2014.

**2. Data and methods**

     We used an existing WRF simulation (cell size: 10 km) for the whole Iberian Peninsula (Figure 1), with a 3 h time step from January 1979 to November 2014, as input data for the FSM. Most inputs of the FSM were extracted directly from the WRF simulation, but some were calculated using other submodels. We projected the WRF

outputs and derived variables to different elevation bands, from 500 to 2900 m a.s.l. at steps of 100 m, from the WRF pixel elevation using several hygrometric and psychrometric formulas and elevation lapse rates. Validation was performed at different steps of the workflow using different observational data sources. Figure 2 shows the workflow completely, which is described in more detail below. Appendix A lists all the

abbreviations used in this study.

  **2.1. Meteorological driving data**

     The meteorological variables were calculated using the WRF model, a mesoscale climate model. Previous researchers used this model to simulate climate at

regional scales for analysis of past, present, and future conditions (Chen et al. 2011; Heikkilä, Sandvik, and Sorteberg 2011). The spatial resolution of our simulation is 0.088º (~10 km) and the time step is 3 h. ERA-Interim reanalysis (Berrisford et al.,



2011) was used as driving data for the WRF model. With this procedure all meteorological variables to run snowpack models were generated for the whole Iberian Peninsula. The WRF configuration was described in detail by García-Valdecasas Ojeda et al. (2017). This simulation provided the following variables: wind speed (Ua), surface temperature (T), precipitation (Pr), relative humidity (RH), short wave incoming radiation (SW), and atmospheric pressure (Ps).

**2.2. Snow energy and mass balance model**

SD and SWE time series were obtained using a mass and energy balance snowpack model. The Factorial Snow Model (FSM) is a multi-physics snow model that simulates the accumulation and melting of snow (Essery, 2015). This model allows selection of two options for parameterizations of five different process, thereby enabling 32 different model configurations. The configuration used to develop our simulations decreases snow albedo and increases snow density at different rates for cold and melting snow, calculates thermal conductivity as a function of snow density, adjusts the turbulent exchange coefficient as a function of the bulk Richardson number, and allows retention and refreezing of liquid water inside the snowpack.

The model works with different numbers and thicknesses of layers, depending on snowpack depth. Thus, it assumes a single layer when snow depth is less than 0.2 m, and a maximum of three layers when the depth is greater than 0.5 m. This configuration allows the model to characterize the highly variable climatological conditions of the Iberian mountains. In addition to the variables provided by the WRF simulation (listed in section 2.1), the FSM also needs estimates of snow rate (Sf), rain rate (Rf), and long wave incoming radiation (LW). To avoid the expense of rerunning WRF in this study, these variables have been reconstructed from available WRF simulation outputs.

To calculate Sf and Rf, it was used a psychrometric energy balance method (PPPm) (Harder and Pomeroy, 2013), which uses relative humidity and air temperature to calculate the surface temperature of falling hydrometeors. From this value, the fraction of liquid precipitation is:

$$f_r(T_i) = \frac{1}{1 + bc^{T_i}} \tag{1}$$

where $fr$ is the percentage of liquid precipitation, $T_i$ is the temperature (ºC) of the falling hydrometeor, and $b$ and $c$ are derived from statistical fits (2.50286 and 0.125006,



respectively, for hourly time intervals). $T_i$ is calculated from Eq. (2), which it was

solved numerically using the method described by Brent (1972):

$$T_i = T_a + \frac{D}{\lambda_t} L \left( \rho_{T_a} - \rho_{sat_{(T_i)}} \right) \qquad (2)$$

where $T_a$ is the temperature (°K), $D$ is the diffusivity of water vapour in air (m$^2$ s$^{-1}$), $\lambda_t$

is the thermal conductivity of air (W m$^{-1}$ K$^{-1}$), $L$ is the latent heat of sublimation or

vaporization (J kg$^{-1}$), and $\rho_{T_a}$ and $\rho_{sat_{(T_i)}}$ (kg m$^{-3}$) are respectively the vapor densities in

free atmosphere and at the saturated hydrometeor surface. This methodology gives the

percentage of liquid precipitation; the percentage of solid precipitation is directly

calculated from $f_r$.

Incoming long wave radiation (W m$^{-2}$) was estimated from the Stefan-

Boltzmann law:

$$L_\downarrow = \varepsilon \sigma T_a^{\ 4} \qquad (3)$$

where $\sigma$ is the Stefan-Boltzmann constant and $\varepsilon$ is the emissivity of the atmosphere.

Emissivity was calculated as a function of elevation and cloud cover, as

proposed by Liston et al. (2006b), who use a variation of the methodology described by

Iziomon et al. (2003). Thus, emissivity is calculated as:

$$\varepsilon = 1.083 (1 + Z_s cc^2)[1 - X_s exp(-Y_s e/T_a)] \qquad (4)$$

where $e$ (Pa) is the atmospheric vapour pressure, $cc$ is de fractional cloud cover and

$X_s, Y_s$ and $Z_s$ are coefficients that are corrected with elevation:

$$
\begin{aligned}
C_S &= C_1 & z < 200 \ m.a.s.l \\
C_S &= C_1 + (z - z_1)\left(\frac{C_2 - C_1}{z_2 - z_1}\right) & 200 \ m.a.s.l \leq z \leq 3000 \ m.a.s.l \qquad (5) \\
C_S &= C_2 & 3000 \ m.a.s.l. < z
\end{aligned}
$$

where $z$ ($m$) is the elevation above sea level, and $X_s, Y_s$ and $Z_s$ can be substituted for $C$,

with $X_1 = 0.35, X_2 = 0.51, Y_1 = 0.100 \ K \ Pa^{-1}, Y_2 = 0.130 \ K \ Pa^{-1}, Z_1 =$

$0.224, Z_2 = 1.100, z_1 = 200 \ m.a.s.l.,$ and $z_2 = 3000 \ m.a.s.l.$

Different parameterizations using SW were tested to estimate $c_c$ ,from potential

SW, a more accurate approach than the parameterization proposed by Liston et al.

(2006b), according to Gascoin et al. (2013). This approach uses the relationship

between SW and potential SW radiation that is restricted to daylight hours. Thus, in this

work, it was used the parametrization proposed by Walcek (1994), which is the original

parametrization proposed by Liston et al. (2006b).



$$c_c = 0.832 exp \left( \frac{RH_{700} - 100}{41.6} \right) \qquad (6)$$

where $RH_{700}$ is the relative humidity at 700 mb.

    The methodology used to project RH to 700 mb elevation is described below. To scale the snow simulations to different elevations, it was first used the internationally accepted standard air temperature lapse-rate ($\beta = 0.0065 \, \mathrm{°} K m^{-1}$) (Barry and Chorley, 1987; ISO, 1975) to project the surface air temperature. For RH, it was used the methodology proposed by Liston et al. (2006b), in which a lapse-rate is applied to the dew point temperature (HRm). First, it was calculated the dew point temperature from RH and the saturation vapor pressure. Then, it was applied the standard air temperature lapse-rate to the dew point temperature, and recalculated the RH at the target elevation from the scaled dew point temperature and the saturation vapor pressure. Once it was rescaled temperature and RH, it was calculated the precipitation phase and LW radiation at the different elevations.

    Finally, to estimate the scaled surface air pressure it was used a generalization of the barometric formula for scenarios that consider air temperature lapse-rates (Bf) (Berberan-Santos et al., 1997):

$$p_{(z)} = p_{(0)} \left( 1 - \frac{\beta * z}{T_a} \right)^{mg/R\beta} \qquad (7)$$

where $p_{(0)}$ is the surface air pressure, $z$ is the elevation difference (m), $m$ is the molecular mass of air (0.0289644 kg mol⁻¹), and $R$ is the universal gas constant (8.31432 J K⁻¹ mol⁻¹).

### 2.3. Validation procedure

    Validation was performed at different resolutions and at different steps of the workflow, using all available observational data (Figure 2). Previous studies (Argüeso et al., 2012; García-Valdecasas-Ojeda et al., 2016) simulated temperature and precipitation using WRF at different time scales compared with the grids, based on observations from Spain02 (Herrera et al., 2012) and PT02 (Belo-Pereira et al., 2011), high-resolution precipitation and temperature gridded datasets for Spain and Portugal, respectively. The results indicated proper simulation of the major patterns of precipitation and temperature, even for extreme events. Subsequent research showed that the downscaling made by WRF provided improved accuracy compared to ERA-Interim data, due to the higher resolution (García-Valdecasas Ojeda et al., 2017).



In this work, it was used the moderate-resolution imaging spectroradiometer
(MODIS) satellite sensor to validate our snow cover product for the period September
2000 to November 2014. Similarly data from telenivometers, which were available in
the Pyrenees from October 2009 to June 2014.

First, it was compared MODIS data with the SD and SWE time series (10 km
resolution). MODIS snow maps were generated using the same workflow to each
mountain range in the study area (Pyrenees, Cantabrian Mountains, Central System,
Iberian Range, and Sierra Nevada). It was downloaded all the available MOD10A1 and
MYD10A1 products (version 5) from the National Snow and Ice Data Center (Hall et
al., 2006). The original granules were mosaicked and re-projected from the sinusoidal
system to the Universal Transverse Mercator (UTM) reference system. Then, it was ran
a gap filling algorithm, using the binary snow product to avoid data losses due to cloud
cover (Gascoin et al., 2015). This provided gap-free daily maps showing the presence
and absence of snow in each mountain range from 2000 to 2014. From these maps, the
probability of snow was calculated as:

$$P_{(Snow)} = \frac{Ns}{N} * 100 \qquad (8)$$

where $P_{(Snow)}$ is the probability of snow (%), $Ns$ is the number of days with snow, and
$Ns$ is the total number of days of the period.

Snow probability maps were also calculated from the FSM snow cover maps. In
this work, it was chosen a threshold of 0.11 m for SD and a threshold of 40 mm for
SWE (Gascoin et al., 2015) in the FSM time series. This allowed to generate snow
cover maps from FSM outputs. Then, it was aggregated the MODIS pixels (500 m) to
the simulation grid (~10 km), with averaging of the values of MODIS pixels to make
them comparable.

It was also used data from 9 telenivometers, which measure sub-hourly SWE
and SD using gamma-ray attenuation and acoustic sensors. These data were provided by
the ERHIN program (Estimación de Recursos Hídricos Procedentes de la Nieve) of the
Hydrological Ebro River Basin Authority (Navarro-Serrano and López-Moreno, 2017).
Eight telenivometers were in the Pyrenees, and one was in the Cantabric Range. It was
also used an SD sensor in the Central System mountain range, which is from the
National Meteorological Agency of Spain (AEMET). It was projected the
meteorological variables from the WRF simulation to elevations of the different





telenivometers for simulations. Figure 3 shows a comparison of the modeled and

observed SD time series at these 10 sites.

It must be noted that it is challenging to validate gridded products from ground-based data (Snauffer et al., 2016). Snowpack can have large variability over small distances (López-Moreno et al., 2015; Meromy et al., 2013). This implies that punctual measurements may not by representative of the 10 km resolution data, even when

comparing a simulation at the same elevation as the telenivometer. In addition, snow measurements always include biases from the different measuring devices (Kinar and Pomeroy, 2015). Thus, we focused on the temporal patterns of snowpack during the season. More specifically, it was compared the accumulation patterns during the season, assuming that accumulation and melting rates were similar in the simulated and

observational data, but that SD and SWE likely differ between the telenivometer and the simulation.

Thus, in this work it was first compared different percentiles of SD and SWE in the telenivometer and the simulated time series. Then, using each percentile as a threshold for snow presence, it was converted the series into binary data, allowing use

of the Kappa test (Cohen, 1960) for each percentile. The Kappa coefficient ranges from 1 and <0, but it is difficult to assign an agreement criterion based on Kappa value. Thus, it was used the thresholds proposed by Landis and Koch (1977), which basically agree with values proposed by Fleiss et al. (1969) (<0.00: poor, 0.00-0.20: slight, 0.21-0.4: fair, 0.41-0.60: moderate, 0.61-0.80: substantial, and 0.81-1.00: almost perfect). Later, it

was examined percentile values between 10% and 90%, as more representative of snow accumulation during the season.

### 3. Results

### 3.1. Validation

Our analysis of the probability of snow presence from MODIS and FSM shows

that the outputs had good correlations (Figure 4). This analysis compared the probability of snow at each pixel (~$10 \times 10$ km) from MODIS and FSM outputs for the SWE and SD time series from September 2000 to November 2014. The mean coefficient ($R^2$) was 0.76, and a mean absolute error was 6.3%. This analysis also shows the correlations for each mountain range, and the distribution of errors for SWE and SD (simulated −

observed).



These results also show there were no significant differences in the errors of $P_{(Snow)}$ for the different mountain ranges. However, the correlation was not strong for the Sierra Nevada range, probably due to its limited snow cover, although this remained inside the variability of the scatterplot.

Validation of these results with telenivometers indicated Kappa values for thresholds in the 10th to 90th percentiles of each season (Figure 5). The Kappa values were mostly above 0.6, although accuracy declined for the highest percentiles.

The Kappa coefficient does not account for the displacement magnitude of the different percentiles, and a difference of a few days in the time of peak accumulation

may cause a sharp decrease in the Kappa value. This is the reason for the loss of accuracy at the highest percentiles. Thus, it was further analyzed these data to determine the time of the year when snowpack exceeded the 90th, 75th, and 50th percentiles at each telenivometer in the observed (OBS) and simulated (SIM) series (Figure 5C). This analysis shows that, despite small temporal shifts, the simulated snow series accurately

represents the temporal patterns when different snow percentiles are exceeded.

The biggest shift in the position of the 90th and 75th percentiles was during the 2011/2012 season. This season was extremely dry on the Iberian Peninsula, and there were very few snowfall events (Figure 3). Thus, a small bias in the simulation of a single event during this time could lead to a large error in prediction of the magnitude

and timing of SD and SWE maxima.

### 3.2. Gridded snow dataset: applications and limitations

The final products of the models are daily gridded datasets (resolution: 0.088º, ~10 km) of SD and SWE at elevations from 500 to 2900 m.a.s.l. (100 m intervals) from 1980 to 2014. The datasets (ncdf-4 format) cover the entire Iberian Peninsula, including

the north side of the Pyrenees, in France. Each dataset contains information of the entire Iberian Peninsula and a mask that covers pixels that do not present areas at the elevations of the simulation estimated from a 250m resolution DEM.

This snow database provides new opportunities for studies of snow in the Iberian Peninsula. In particular, the temporal resolution and the duration of the series are

significant improvements over previous observational data. Also, the geographic data on SD and SWE generated provides the opportunity to obtain more snow and hydrologically relevant information than that available from remote sensing alone. It is



also possible to develop different snow products at different elevations, allowing comparison of different elevations and different regions. For example,

Figure 7 shows examples of other snow variables that can be derived from the database: average number of snowfalls and percentage of days with snow cover at three elevations. These analyses are particularly useful for the development of different snow climatologies for the whole Iberian Peninsula, or for specific areas, in studies that rely on ecological data (*e.g.* phenology or distribution of plants and animals, forest growth,

etc.), studies that require hydrological parameters for different catchments, and studies that determine risk maps for snow-related events.

It is also possible to extract daily time series for different areas or elevations at each pixel. For example, Figure 8 compares SWE series at three elevations in the pixel at the highest peak of the Pyrenees (Aneto Peak, 3404 m.a.s.l.). Thus, these series allow

study of different annual snow accumulation and melting patterns on a specific location and how elevation influence snow evolution. Similarly, it enables to study the existence of temporal trends or the occurrence of extreme snowfall and melting events.

The database contains uncertainties that were not easy to quantify, due to the limited amount of observational data. Biases may be due to uncertainty of the boundary

conditions from the ERA interim reanalysis (Chaudhuri et al., 2013) since errors from the WRF downscaling model are difficult to quantify in mountain areas (Gutmann et al., 2012), and uncertainties that typically result from simulations of snow mass and energy balance from meteorological data (Essery et al., 1999, 2013; Magnusson et al., 2015). The use of the standard air temperature lapse-rate can be also a source of uncertainty.

Despite other studies have observed a decrease in the lapse-rate during winter months, this effect is result of thermic inversions that are not considered due to the spatial resolution of the simulation.

Despite these limitations, we had very satisfactory results when testing the duration and the interannual variability of the snowpack against MODIS and

telenivometer data, which provided reliable observations during several snow seasons. This way, the database presents a reliable validation for more than a third of the time period generated. When using this database, it is important to consider that it was based on the assumption of flat topography within each $10 \times 10$ km pixel. Therefore, this dataset is not suitable for studies of snow variability due to terrain aspect, slope, and

snow redistribution processes, such as avalanches and wind transport.



### 4. Conclusions

It was presented a new daily gridded database of SD and SWE for the Iberian
Peninsula from 1980 to 2014 period at a resolution of 0.088º (~10 km). The database
consists of 50 ncfd-4 files for SD and SWE from 500 to 2900 m.a.s.l., and another 2 at
WRF simulations DEM, summing more than 652,000 maps. A mask label as "no data"
is included if the grid is not found at the elevation of the simulated elevation band.

The scarcity of snow observations in the Iberian Peninsula made it necessary to
couple a dynamical downscaling of Era-Interim reanalysis using the WRF model by use
of a snow energy and mass balance model (FSM). Input data of FSM provided directly,
or estimated from WRF outputs, were available for the average elevation of each 10 ×
10 km pixel, and these data were transformed to achieve an elevation offset at 100 m
intervals.

Despite some uncertainties, the database is consistent with available
observational data. More specifically, validation with MODIS data indicated an error of
6.07% and an $R^2$ of 0.76 in analysis of the mean presence of snow. The database also
provides good representation of the temporal patterns of the telenivometers, with Kappa
values generally over 0.6, and above 0.4 for all analyzed percentiles.

This database will be an important resource for studies of many different
hydrological, environmental, and economic processes in Mediterranean areas. Thus, we
expect the database presented here will be useful for future snow-related studies at
regional scales in the Iberian Peninsula, and for a broad community of researchers and
land managers working in areas where snow occurs.

### 5. Data Availability

The data presented here are available for free download from Zenodo
(https://zenodo.org/record/854619). SD and SWE datasets are in ncdf4 format, with one
file for each elevation band. The observational information used to validate the main
data is also available for download. All telenivometer data are in .csv format. Daily
snow cover (derived from MODIS) is provided as 5 multiband GeoTiff files (one file
for each mountain range, each band is a date), and a .csv file indicates the date of each
band.
The FSM code is freely available from https://github.com/RichardEssery/FSM





## Appendix A: Variable summary

Models and elevation correction technics:

| Name | Acronym | Function |
| --- | --- | --- |
| Weather research and forecasting | WRF | Regional Climate Modelling |
| Factorial snow model | FSM | Snowpack Modelling |
| Relative humidity projection | HRm | Project the relative humidity |
| Barometric formula | Bf | Project the atmospheric pressure |
| Precipitation-phase partitioning | PPPm | Divide precipitation phase (Snow/Rain) |
| Long wave model | LWm | Estimate Long wave incoming radiation |

Variables:

| Name | Acronym |
| --- | --- |
| Wind speed | Ua |
| Precipitation | Pr |
| Temperature | T |
| Relative humidity | RH |
| Atmospheric pressure | Ps |
| Short wave incoming radiation | SW |
| Long wave incoming radiation | LW |
| Snowfall rate | Sf |
| Rainfall rate | Rf |
| Snow depth | SD |
| Snow water equivalent | SWE |

**Acknowledgments:**

Esteban Alonso-González is granted by the Spanish Ministry of Economy and Competitiveness (BES-2015-071466).This study was funded by the Spanish Ministry of Economy and Competitiveness projects CGL2014-52599-P "*Estudio del manto de nieve en la montaña española y su respuesta a la variabilidad y*
*cambio climatico*" and (with additional support from the European Community funds (FEDER)) CGL2013-48539-R "*Impactos del cambio climático en los recursos hídricos de la cuenca del Duero a alta resolución*.". Also the Regional Government of Andalusia has found this research with the project P11-RNM-7941 "*Impactos del Cambio Climático en la cuenca del Guadalquivir (LICUA)*". The authors would like to express his thanks to Hydrological Ebro River Basin Authority (CHE) for
providing telenivometers data. Development of FSM is supported by NERC grant NE/P011926/1

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

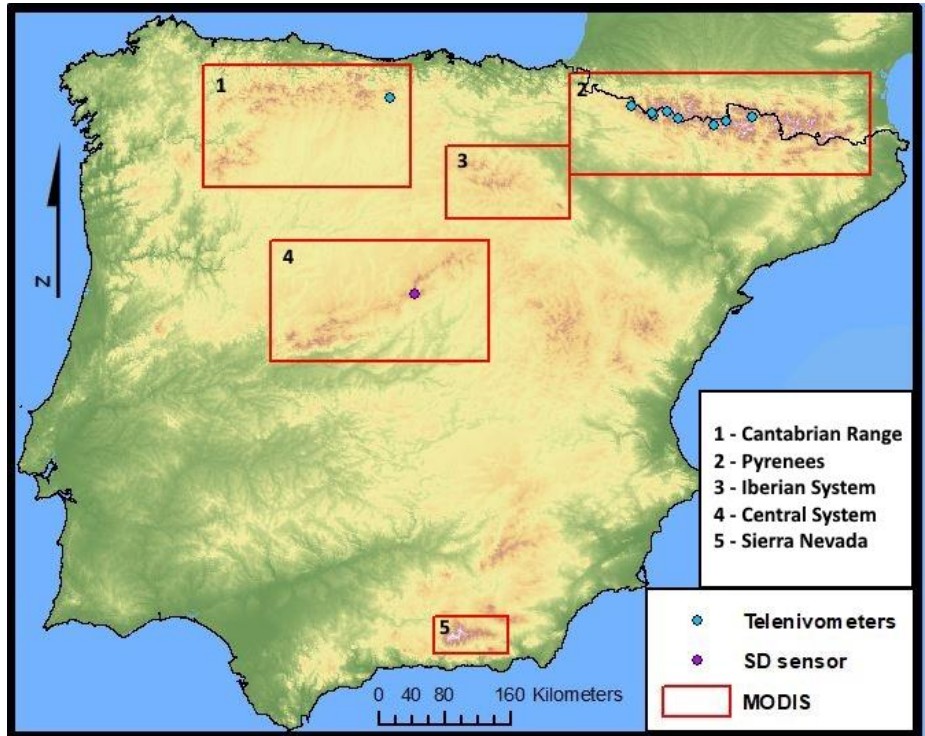

**Figure 1: Digital elevation model of the Iberian Peninsula and locations of the telenivometers, Cotos Pass SD sensor and MODIS study areas.**

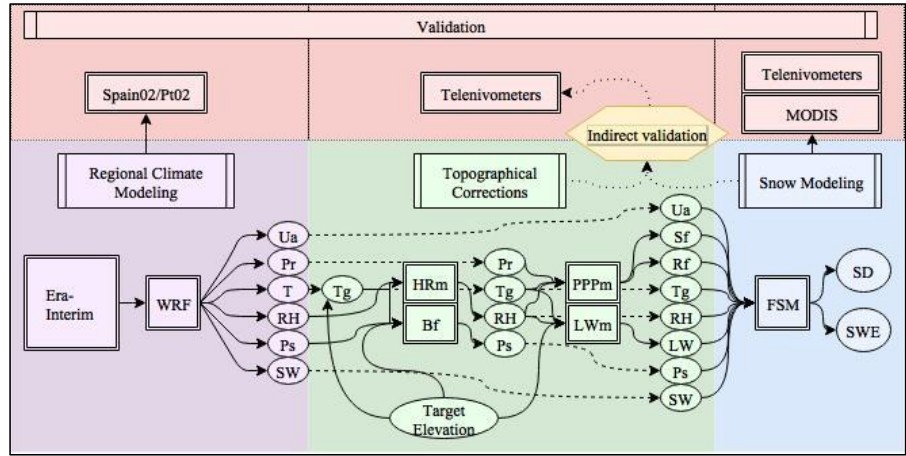

**Figure 2: Simulation workflow. Squared boxes represent modelling steps and rounded boxes represent meteorological variables. Variables that are not inputs or outputs of a model are indicated by dotted lines (see a glossary of used abbreviations in Appendix A).**



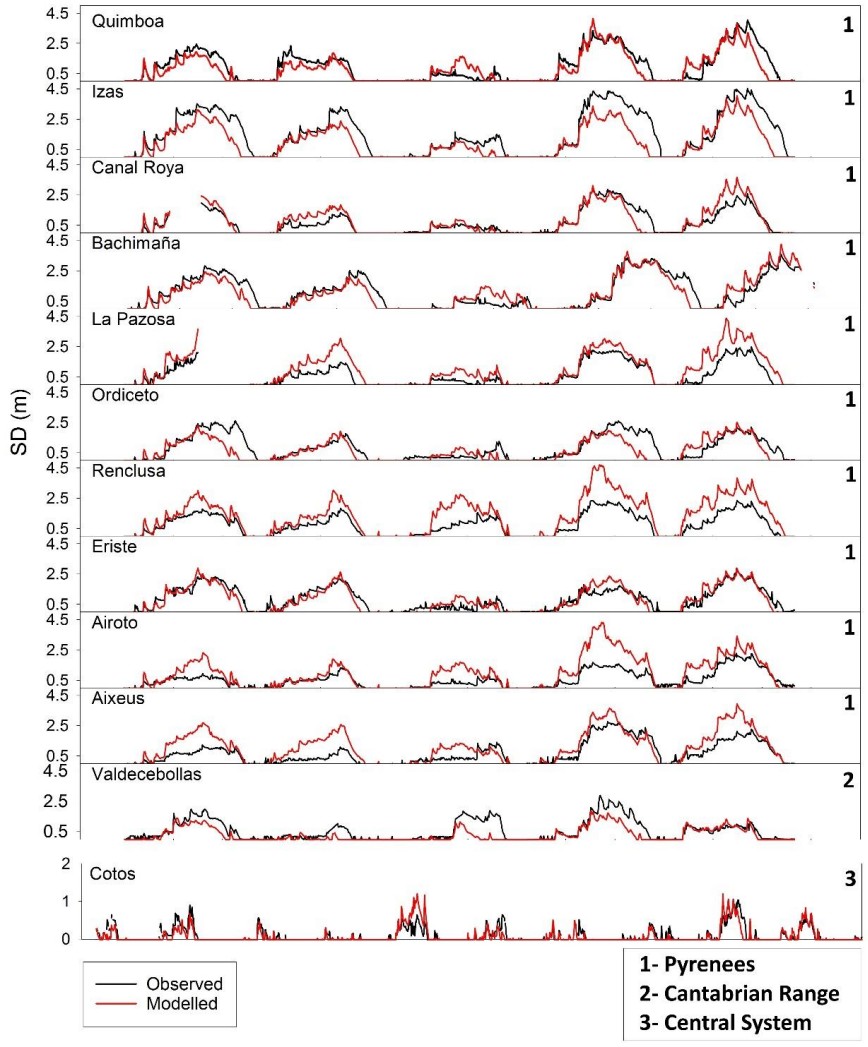


**Figure 3: Comparison between modeled (red) and observed (black) SD time series for each telenivometer and the Cotos SD sensor.**



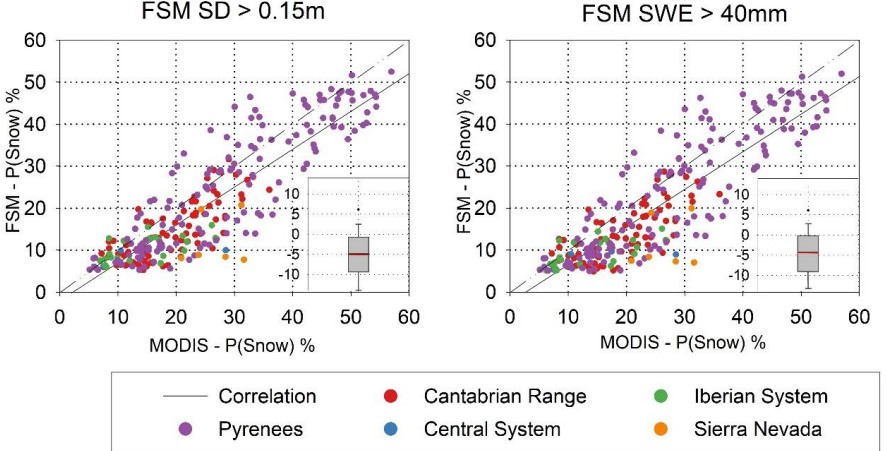

**Figure 4: Correlation between the long-term (2000-2015) mean probability of snow depth (left) and snow water equivalent (right) from MODIS data and from FSM output. Box plot insets show the frequency distributions of errors (%), with the central red lines indicating average errors, boxes indicating the 25th and 75th percentiles, bars indicating the 10th and 90th percentiles, and dots indicating the 5th and 95th percentiles.**

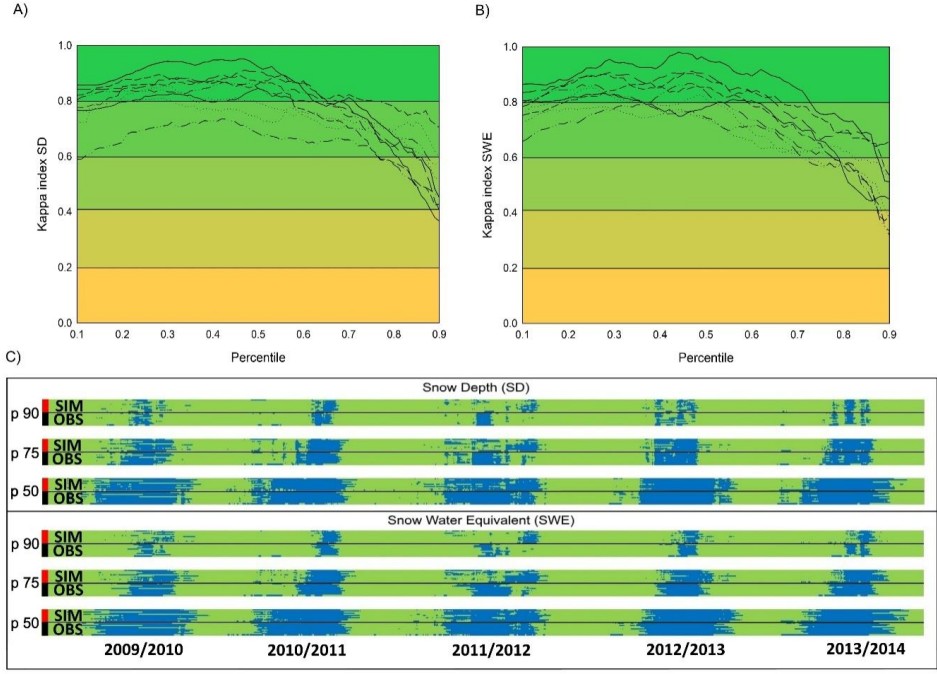

**Figure 5: Kappa values derived from comparison of observed and simulated series for different percentiles of snow depth (A) and snow water equivalent (B), and periods of the year (blue) when snowpack exceeds the 90th, 75th, and 50th**



percentiles (C). In C, each pair of bands show the times when the different
percentiles in the observed (OBS) and simulated (SIM) series at each telenivometer
exceeded the indicated percentile.

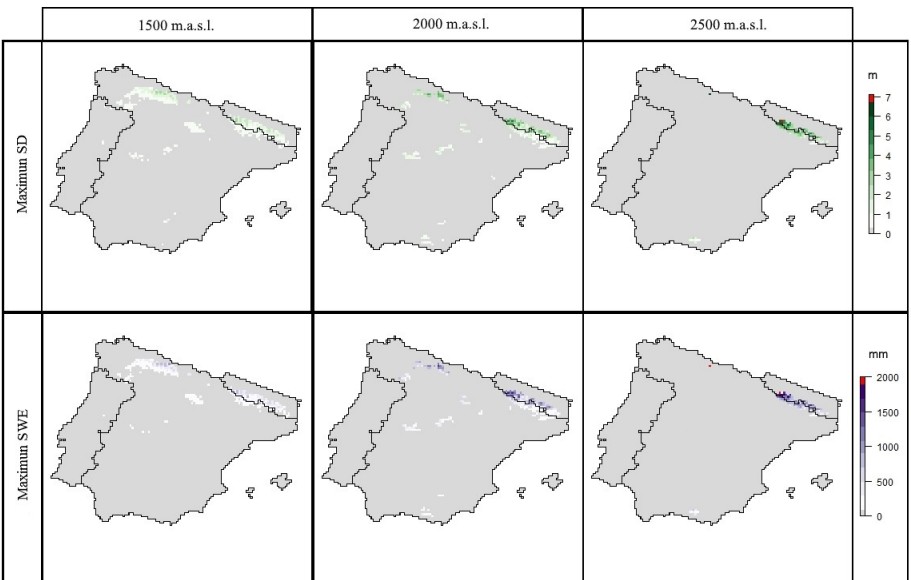

**Figure 6: Long-term (1980-2014) average maximum SWE and SD grids at 1500,
2000, and 2500 m.a.s.l.**

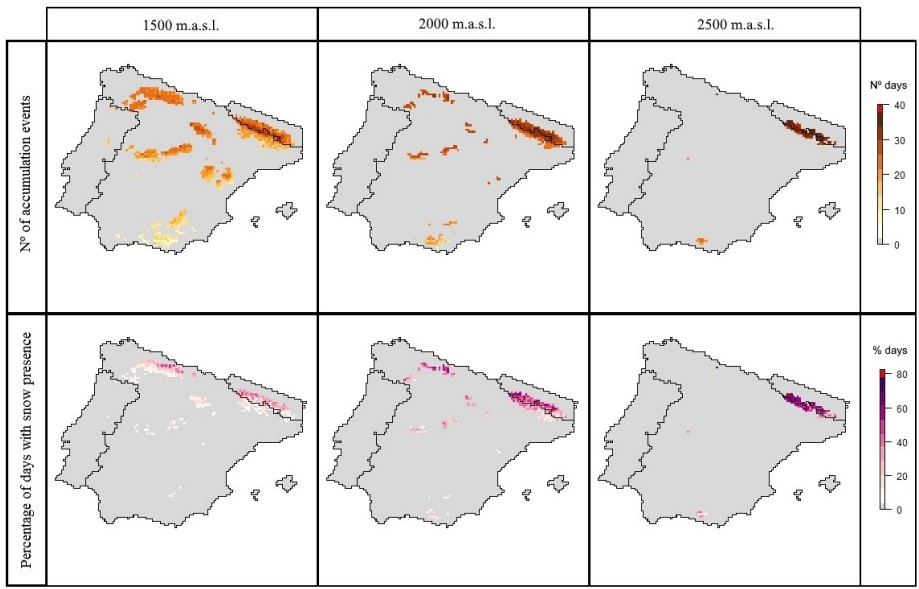


**Figure 7: Long-term (1980-2014) average number of snowfall events and
percentage of snow presence at 1500, 2000 and 2500 m.a.s.l.**



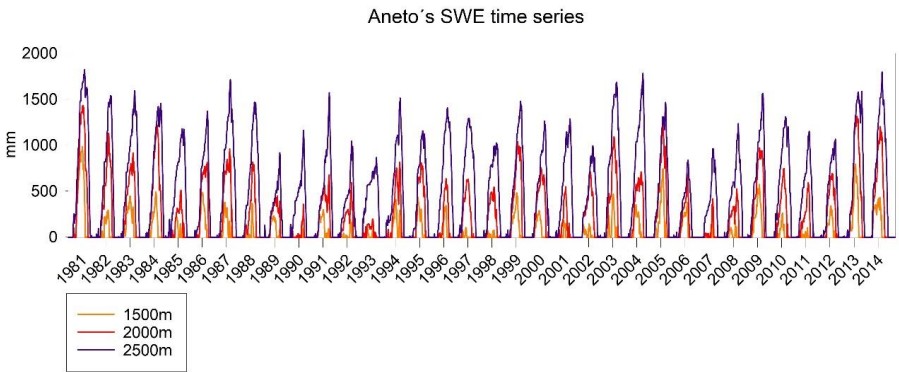

**Figure 8: Comparison of SWE time series at 1500, 2000, and 2500 m.a.s.l. at Aneto Peak.**
