# Peer review of "Daily gridded datasets of snow depth and snow water equivalent for the Iberian"

_Earth System Science Data, 2017_

## Referee Comment (RC1) · J. Herrero (Referee) · 24 Nov 2017

**General Comments**

This paper presents us an interesting dataset of estimated snow depth and water equivalent for the Iberian Peninsula. It was obtained by a laudable combination of meteorological reanalysis, regional climate modelling, and detailed snow modelling. The dataset is relevant and worthy of publication. The methodology is flawless, interesting, and well presented.

**Specific comments**

[Figure]

- **The workflow used in this paper, from reanalysis to modelling, is really meritorious. Several hypothesis and parameterizations had to be used to obtain some inputs required for the snow modelling (lapse rates, cloud cover, fraction of solid precipitation...). They make a nice chain of hypothesis but, at the same time, add a lot of uncertainty to the input data derived from them. The authors themselves recognize, rightly, these limitations (section 3.2 and line 338 and following) and have clearly stated that the dataset may be useful only at regional scale studies. Besides, the resolution of the simulation, despite the usage of bands for the solid fraction of precipitation, is in the order of magnitude of the size or width of the mountain ranges we are dealing with in Spain, with the exception of the Pyrenees. I think that in most of these areas this is an important drawback to use this dataset, not only for avalanches or wind-driven phenomena, but also for hydrological or environmental applications (Line 373-374), which are here very dependent on local constrictions. I guess that the use of a more detailed model is feasible and a good next step. This is not a question in itself, but rather a comment to generate some discussion on this topic.**

- **Did you carry out any kind of calibration of the snow model FSM? What criteria have been used to decide the configuration and the value of the parameters of the model?**

- **Even though English is not my mother tongue, I think there are many poorly constructed sentences in what seems like a bad translation from Spanish passive tense. The paper would require a thorough grammatical revision. Examples:**

    - **Line 201. Original: 'For RH, it was used the methodology proposed by...'. Suggestion: 'For RH, the methodology proposed by... was used.'**
    - **Line 203. 'First, it was calculated the dew point temperature from...' by**

[Figure]

'First, the dew point temperature was calculated from...'

– Line 234. 'It was downloaded all the available...' by 'All the available ...were downloaded'

– Line 248. 'Then, it was aggregated the...' by 'Then, the MODIS pixels (500 m) were aggregated to...'

– Line 257. 'It was projected the meteorological variables from...' by 'The meteorological variables were projected from...'

– Line 274. 'It was converted the series into... ' by 'The series were converted into...'

– And so on.

**Technical corrections**

• Line 40: extra comma between 'economy' and 'of'

• Line 181 'de' by 'the'

• Line 194. 'parametrization proposed by Walcek'... Do you mean parametrization of Cc or SW?

• Line 229-230. The sentence seems incomplete.

• Line 232. 'Same workflow to each...' by 'Same workflow for each...'

• Lines 517 and 520. The author enumeration is duplicated (Liston and Elder).

• Lines 556-557 The same as above. And something similar in other references, like Line402-403, Line 425, Line 429, Line 482, Line 507, Line 536, Line 569-575 (this is specially funny... more than 100 authors!), Line 603,

• Line 448. '?' in place of a missing character

**Interactive comment on Earth Syst. Sci. Data Discuss., https://doi.org/10.5194/essd-2017-106, 2017.**

---

## Referee Comment (RC2) · Dr. Santos (Referee) · 7 Dec 2017

"**Daily gridded datasets of snow depth and snow water equivalent for the Iberian Peninsula from 1980 to 2014**" by Alonso-González *et al.*

**General Comments:**

Based on Earth System Science Data's review criteria I consider that the paper presented by Alonso *et al.*, is of great interest to the scientific community, particularly to the ones working in the cryospheric field. Therefore, this paper and the respective dataset is significant, useful and worthy of publication.

The methods used in this work are not entirely new, but they have been appropriately adjusted to the Iberian Peninsula study areas where this methodology has not been applied so far, at least not at the extent presented in this study. The authors show a remarkable effort to combine several sources of input data as well as to create/simulate new data. Thus, I consider that this data set presented for the Iberian mountains is very interesting and unique to a great extent.

Regarding the data quality, I confirm that the data is easily accessible and readable. Concerning the presentation quality, the paper is quite clear and is not too long.

**Specific comments:**

- The reanalysis and modelling of Snow Depth (SD) and Snow Water Equivalent (SWE) for the Iberian Peninsula mountains presented in this paper is of great interest as mentioned before. However, and as the authors have rightly pointed, there are some limitations related with the applicability of the dataset. From my point of view and experience in snow variability in Sierra Nevada, the coarse resolution of the results (10 Km) would make difficult to use the data set for hydrological or risk management studies, as this topics require a much more detailed approach. However the utility of the data set generated and presented in this paper is not dubious or doubtful, but it should be considered for larger scale analysis and not for local studies.

  In this regard, would you consider in the future to use a combination of MODIS images (good temporal resolution) with Landsat or Sentinel imagery (better spatial resolution) (lines 227-229)?

- Concerning the snow energy and mass model balance model, is not clear to me if parameters like emissivity are estimated daily (line 176-179) or hourly (as explained in line 164)? This should be clarified.

- For Sierra Nevada there are available DEM at a better spatial resolution. This is just a suggestion that could help you to improve the quality of the results.

- Regarding the grammatical revision and technical corrections, I would suggest the revision of the same aspects already pointed by other referee. So in this regard I would only add two corrections/suggestions:

- Line 244: The authors duplicate de *Ns* parameter. I guess the second one should be only *N* refereeing to the total number of days of the period.

- On the figure 1 (page 20), I would probably add a small table with some details about the Telenivometers and the SD sensors (mountain range, location, altitude, orientation…). Also on figure 3 (page 21) there are 10 different locations with Telenivometers but on figure 1 we can identify only 8-9 Telenivometers. Maybe the Telenivometers location is so close to each other (in some cases) that the symbols are overlapped in map (Fig. 1)?

Finally I would like to greet the authors for this interesting study.

---

## Author Comment (AC1) · 28 Dec 2017

The comment was uploaded in the form of a supplement:
https://www.earth-syst-sci-data-discuss.net/essd-2017-106/essd-2017-106-AC1-supplement.zip
* * *